# Impact of Basalt Filler on Thermal and Mechanical Properties, as Well as Fire Hazard, of Silicone Rubber Composites, Including Ceramizable Composites

**DOI:** 10.3390/ma12152432

**Published:** 2019-07-30

**Authors:** Przemysław Rybiński, Bartłomiej Syrek, Witold Żukowski, Dariusz Bradło, Mateusz Imiela, Rafał Anyszka, Anke Blume, Wouter Verbouwe

**Affiliations:** 1Institute of Chemistry, The Jan Kochanowski University, Żeromskiego 5, 25-369 Kielce, Poland; 2Department of Chemical Engineering and Technology, Cracow University of Technology, Warszawska 24, 31-155 Kraków, Poland; 3Institute of Polymer & Dye Technology, Faculty of Chemistry, Lodz University of Technology, Stefanowskiego 12/16, 90-924 Łódź, Poland; 4Department of Mechanics of Solids, Surfaces & Systems (MS3), Faculty of Engineering Technology, Chair of Elastomer Technology & Engineering, University of Twente, 7500 AE Enschede, The Netherlands; 5Basaltex NV, Zuidstraat 18-8560 Wevelgem, Belgium

**Keywords:** basalt flake, basalt fibers, composite, silicone rubber, ceramization (ceramification), ceramizable composite, fire hazard, thermal properties, mechanical properties

## Abstract

This article illustrates the impact of basalt filler, both in the form of basalt flakes and basalt fibers, on thermal and mechanical properties, as well as on the fire hazard, of silicone rubber (SR) composites, including ceramizable composites. In addition to basalt filler, ceramizable composites contain mineral fillers in their composition in the form of silica and calcium carbonate, inorganic fluxes such as zinc borate and glass frit, and melamine cyanurate as a flame retardant. The obtained composites were analyzed from the point of view of their morphology, rheological and thermal properties, flammability, and mechanical properties before and after the ceramization process. The obtained research results indicate that the basalt filler has an unambiguous impact on the improvement of thermal properties and the reduction of flammability in the analyzed composites. The results of morphological analyses of ceramizable composites before and after the process of their ceramization indicate a definite impact of the basalt filler on the structure of the formed ceramic layer. An increase in its homogeneity exerts a direct impact on the improvement of its mechanical parameters.

## 1. Introduction

Ceramizable composites are new polymer materials that are either non-flammable or characterized by a reduced flammability and are intended, above all, to protect copper wires in electrical cables against melting resulting from fire or high temperature by forming a protective ceramic layer. During regular usage, they are characterized by properties similar to conventional elastomer composites, whereas in the case of fire they are transformed into an insulating ceramic layer of a specific porosity and mechanical durability, which ensures the integrity of the electrical circuits for a minimum period of 120 min from the point at which flame appears. Although ceramizable materials may be obtained through the application of various polymers, e.g., EVA [1,2,3], PE [4], SBR [5,6], NBR [7], EPDM [8,9] etc., silicone rubber is considered to be the optimal precursor [10,11] due to the high energy value of the Si-O bond in the backbone chain.

The review of the source literature on the subject leads to the conclusion that the high resistance of silicones to fire is mainly related to the formation, during their thermal decomposition, of a boundary layer composed of silica with a high thermal capacity, which to a considerable degree makes conducting heat between the flame and the sample difficult. On the other hand, the cohesive forces between the particles of silica are insufficient for the formation of an effective barrier impeding the mass flow between the sample and the flame [12,13]. Hence, the formation of an insulating ceramic layer of a specific mechanical durability and which is resitant to cracking requires the introduction of particular mineral fillers into a silicone rubber SR (silicone rubber) matrix.

There are numerous mechanisms responsible for the ceramization process. The most important ones include:

Formation of physical connections between thermally stable particles of mineral fillers, which are performed by low-temperature fluxes, such as: glass frits, which are characterized by a melting point at a level of 374–525 °C [14,15], boron oxide with a melting point of 450 °C, or zinc borate with a melting point of 550 °C.

Formation of silicate bridges between the particles of silica, resulting, among other things, from the decomposition of silicone rubber, and other mineral fillers introduced into the SR matrix. In this mechanism, the key role is played by a good adhesion between silica particles and thermally stable particles of mineral fillers.

Sintering of mineral filler particles occurs through the condensation of hydroxyl groups present on their surface. This mechanism takes place when the content of the mineral fillers in the SR rubber matrix is high.

“In situ” formation of new mineral phases as a result of chemical reactions takes place between the particles of mineral fillers. This mechanism can be exemplified by the reaction between calcium oxide and silica, leading to the formation of calcium silicates, which positively influence the mechanical parameters of the formed ceramic structure [16].

The basic problem that limits the industrial application of ceramizable composites is not only their reduced resistance to fire and elevated temperatures, but also the insufficient mechanical parameters (mainly compression strength) of the formed ceramic structure. It can be seen in the source literature that the existing research works were intended to both increase the fire resistance of ceramizable composites and to improve the mechanical parameters of the ceramic layer formed as a result of their thermal decomposition [17].

From the literature review, it follows that Hu et al., when analyzing the impact of the combination ammonium polyphosphate/aluminum hydroxide/mica, stated that the decomposition of APP (ammonium polyphosphate) and Al(OH)_3_ takes place within the temperature range of 300–600 °C, forming new products, i.e., AlPO_4_ and NH_4_AlP_2_O_7_, which, when subsequently reacted with mica at a temperature of 800 °C, are responsible for the formation of KAlP_2_O_3_ and Al_2_O_3_ × 2SiO_2_. An increase in the temperature above 1000 °C leads to the formation of KAlSi_3_O_8_, which imparts a specific bending resistance to the formed ceramic layer [18]. Lou et al. analyzed the impact of both dust and glass fiber on the properties of the ceramic layer [19]. In our previous studies, we carried out research on the impact of carbon fiber on the properties of ceramizable composites of silicone rubber [20]. We also examined the impact of various mineral fillers, including organically modified montmorillonites, on the properties of ceramizable composites of the SR rubber [21].

The present article illustrates the results of research regarding the impact of the basalt filler, both in the form of basalt flakes and basalt fiber, on the properties of silicone rubber ceramizable composites.

The literature on the subject comprises a few papers regarding the impact of the basalt fiber on the properties of epoxy resin [22,23] or polyethylene [24], but currently there have been no publications of results of the research works regarding ceramizable composites containing basalt filler. The research results presented in this article should therefore be treated as pioneer ones when it comes to the improvement of the properties of ceramizable elastomer composites with respect to thermal stability, fire resistance and ceramization performance.

## 2. Materials and Methods

### 2.1. Materials

The polymer matrix was methylvinylsilicone rubber (SR), made by Silikony Polskie (Nowa Sarzyna, Poland) with a vinyl group content of 0.05–0.09% (mol × mole^−1^). The rubber was cross-linked with the use of dicumyl peroxide from Sigma Aldrich (Schnelldorf, Germany) in a quantity of 2.5 parts by wt./100 parts by wt. of the rubber.

The following compounds were used as fillers in the rubber blends: Aerosil 200 fumed silica (BET specific surface area 175–225 m^2^/g; mass loss on drying ≤1.5%, obtained from Evonik Industries (Wesseling, Germany)); melamine cyanurate (MC) (trade name Evermel Aktive 47, obtained from Everkem, Milano, Italy); calcium carbonate anhydrous, free-flowing, reagent ≥99% (obtained from Sigma-Aldrich, Schnelldorf, Germany). The ceramization-promoting glass frits “A 2120” with chemical composition (wt.%): 7.3CaO-25Na_2_O-2.1K_2_O-4.2Al_2_O_3_-61.4SiO_2_ and a softening point temperature of 540 °C was obtained from from Reimbold & Strick GmbH, Cologne, Germany; zinc borate ZnO ≥ 45%, B_2_O ≥ 36% was obtained from Sigma-Aldrich (Essen, Germany); basalt flakes (BLF) and basalt fibers (BFS) (trade name BCS13-6.35-DRY length 0.02 m; width 0.02 m; height 0.022 m, obtained from Basaltex (Wevelgem, Belgium)) (Figure 1).

### 2.2. Preparation of the Samples

The composite mixes (Table 1 and Table 2) were prepared using a laboratory two-roll mill (the rolls had length 200 mm, diameter 150 mm, and were sourced from Bridge, City, UK), working with a rotation speed of 18 rpm (revolutions per minute) for the slower roll and 20 rpm for the faster roll (friction 1:1). The kinetics of vulcanization of the composite mixes were tested by means of a rheometer (Alpha Technologies MDR2000 (Alpha Technologies, Hudson, OH, USA)) according to the ISO 37:1994 standard (temperature: 160 °C, frequency: 1.667 Hz, strain: 0.5°). According to the results, the samples were shaped and vulcanized in steel molds by a laboratory press at 160 °C under 10 MPa of pressure.

### 2.3. Techniques

Photographs of BFL and BFS were obtained through the use of a TPL Trino stereoscopic microscope equipped with a DLT-Cam PRO 5 MP camera and a scanning electron microscope (SEM), model TM3000 from Hitachi (Fukuoka, Japan). 

The thermal properties of the rubber composites were tested under air atmosphere at temperatures ranging from 25 to 700 °C with the use of a thermal analyzer (Jupiter STA 449F3, Netzsch Company, Selb, Germany). The weighted portions amounted to about 5–10 mg. The investigated samples were analyzed at a heating rate of 10 °C min^−1^.

The flammability of the composites was determined using a cone calorimeter (Fire Testing Technology Ltd., East Grinstead, UK). Composite samples with dimensions of (100 × 100 ± 1) mm and thicknesses of (2 ± 0.5) mm were tested in a horizontal position with a heat radiant flux density of 35 kWm^−2^. During the tests, the following parameters were recorded: initial sample weight, time to ignition (TTI), sample weight during testing, total heat released (THR), effective combustion heat (EHC), average weight loss rate (MLR), heat release rate (HRR), final sample weight.

Ceramization of the vulcanizates was performed in a laboratory furnace by Nabertherm with the P320 controller (Lilienthal, Germany). Cylindrical samples (diameter—16 mm; height—8 mm) of the ceramizable composites were heated in 3 different conditions: (1) 1100 °C—from room temperature to 1100 °C in 30 min (heating rate 35 °C/min); (2) 950 °C—from room temperature to 950 °C in 120 min (heating rate 7,5 °C/min), and (3) 550–1000 °C—from room temperature to 550 °C in 53 min (heating rate 10 °C/min), 10 min of isothermal conditions at 550 °C, and end heating from 550 to 1000 °C in 27 min at the (heating rate 16 °C/min)—total time 90 min.

The micromorphology of the composites was examined before and after ceramization by means of scanning electron microscopy (SEM) Joel JSM-6400 (JEOL Ltd., Tokyo, Japan) with Noran energy-dispersive X-ray spectroscope unit (EDS) (Thermo Fisher Scientific, Waltham, MA, USA). The sample cross-sections were prepared by crushing in liquid nitrogen and gold sputtering directly before measurement.

The mechanical properties of the composites were tested by means of a Zwick/Roell 1435 static testing machine (Ulm, Germany), measuring stress at different degrees of elongation (SE100, SE200 and SE300), tensile strengths (Ts), and elongations at break (Eb).

The compression strength of the composites after their ceramization (cylindrical samples: diameter—16 mm; height—8 mm) was tested in the diameter direction by means of a Zwick/Roell Z 2.5 device.

## 3. Results and Discussion

### 3.1. Kinetics of Composites

The introduction of the basalt filler, both in the form of basalt flakes (BFL) and basalt fiber (BFS) has practically no influence on the values of scorch time of the SR rubber composites. On the basis of the data shown in Table 3, it should also be noted that BFL does not influence the values of the torque (Torque at t_05_ and t_09_) of the SR composites, including the ceramizable composites.

The high value of torque at t_09_ of the SR-4 compound, with reference to the other samples, is related to the specific behavior of the rubber and the long fibers during testing. At the beginning of the kinetic of vulcanization test, only uncured viscous rubber responds to the externals stress applied by the rheometer. Uncured rubber of low viscosity moves easily around the large fibers, which do not take part in carrying stress. When the rubber starts to cure, it is no longer able to flow viscously, and starts to transfer the stress to the long and stiff fibers, resulting in a significant increase of torque. This effect gives a much higher torque value than the hydrodynamic effect of the dispersed particulate fillers (Table 3, Figure 2).

### 3.2. Thermal Properties of Composites

Cross-linked silicone rubber SR (sample SR-0) undergoes a clear two-step thermal decomposition. The first step, which begins at a temperature of T = 335 °C, and whose maximum rate of thermal decomposition at a temperature of T = 383 °C amounts to 0.98%/min, is related above all to the reactions of depolymerization and degradation of SR rubber to low-molecular-weight cyclic volatile compounds. The second step of thermal decomposition, whose beginning is registered at a temperature of T = 420 °C, and whose maximum rate at a temperature of T = 516 °C amounts to 7.15%/min, is related to the reactions of both degradation and thermal destruction (Figure 3) [13,25,26].

The radical mechanism constitutes the dominant mechanism of decomposition of the cross-linked silicone rubber. An increase in temperature results in homolytic random dissociation of the Si-C bonds in the backbone polymer chain, as a consequence of which the resulting macroradicals, undergoing the subsequent reactions of recombination and cross-linking, reduce the elasticity of the polysiloxane chain, and hence the efficiency of formation of low-molecular-weight cyclic compounds, with reference to the first step of thermal decomposition.

Both the first and the second step of thermal decomposition of the cross-linked SR rubber are accompanied by a strong exothermic effect registered on the curve DSC (Figure 3). The introduction of the basalt filler into the SR rubber matrix, both in the form of basalt flakes (BFL) and basalt fiber (BFS), does not alter the nature of the thermal transformations; however, it does significantly influence the values of the parameters of thermal indicators (Table 4, Figure 4).

It should be clearly emphasized that only 15 parts by weight of the basalt filler definitely increases thermal stability of the cross-linked SR, expressed both by the temperature of the sample at 5 and 50% mass loss, and by the maximum temperature of the first and second steps of thermal decomposition, the maximum rate of the sample mass loss, and the residue after thermal decomposition at a temperature of T = 650 °C (Figure 4, Table 4). The thermal stability of the composites containing basalt flakes increases proportionally with the increase in the content of basalt flakes in the SR rubber matrix. It should be noted that with the content of the basalt filler above 22.5 parts by weight, the decomposition of the analyzed composites does not exceed 50% of the sample mass. With an increased basalt filler in the SR rubber matrix, the rate of thermal decomposition of the analyzed composites decreases—the parameter dm/dt (Table 4). This indicator is extremely important from the point of view of the fire hazard posed by the examined composites, which is directly related to the formation of smaller amounts of volatile—including flammable ones—products of pyrolysis reaching the flame.

It is also worth noting that, together with an increasing content of the basalt filler in the composite, the value of the parameter P_650_ also increases. Thermally stable basalt, not undergoing any thermal transformations, positively influences the structure of the boundary layer which is formed during the thermal decomposition and combustion of the analyzed composites, and which efficiently impedes the flow of mass and energy between the sample and the flame. A higher value of the parameter P_650_ of the composite SR-4, which contains basalt fiber, with reference to the composite SR-3 containing the same number of basalt flakes, is due to a larger size, and hence worse distribution of BFS in relation to BFL in the polymer matrix.

The introduction of mineral fillers into the silicone rubber matrix—ceramizable composites SR-5÷SR-9—does not, in principle, change the nature of thermal decomposition of the SR rubber (Figure 5).

It should, however, be clearly emphasized that the thermal effect related to the first step of thermal decomposition changes. A high thermal capacity of mineral fillers causes the nature of the thermal effect accompanying the first step of thermal decomposition to change from *exo-* to *endo*thermic (Figure 3). In the presence of mineral fillers, an impact of the basalt filler on the value of the parameters T_5_, T_rmax1_ and T_rmax2_ is practically unnoticeable. However, the basalt filler, both in the form of BFL and BFS, when also in the presence of mineral fillers, influences the values of the parameters dm/dt and P_650_, which proves that it effectively inhibits the processes of thermal decomposition and combustion in the analyzed ceramizable composites of SR rubber.

### 3.3. Flammability

The research results obtained by applying the cone calorimeter method clearly demonstrate that the basalt filler, both in the form of basalt flakes (BFL) and basalt fiber (BFS), clearly reduces the fire hazard of those vulcanizates that contain them. The introduction of only 15 parts by weight of basalt flakes into the silicone rubber matrix leads to an increase in the time to ignition from 115 to 145 s. Increasing the concentration of the basalt filler to 20 parts by weight, both in the form of basalt flakes and basalt fiber, extends the time to as much as 190 s. An increase in the value of the parameter t_i_ (time to ignition) is accompanied by radical extension of the time to flameout (parameter t_f-o_), which indirectly confirms the intensity of the combustion process of a given material (Table 5).

The parameter HRR_max_ (maximum heat release rate) is a key magnitude illustrating the rate of thermal decomposition and combustion of the marked composites. It should be clearly noted that the introduction of 15 parts by weight of BFL into the silicone rubber matrix results in a reduction of the value of the parameter HRR_max_ from 134.7 kW/m^2^ to 96.2 kW/m^2^. An increase in the content of BFL in the SR matrix to 20 parts by weight causes a reduction of the value of the parameter HRR_max_ by as much as 63.9%, i.e., to a value 48.5 kW/m^2^ (Table 5, Figure 6).

A high flame retardant efficiency of the basalt filler is also confirmed by fire hazard parameters such as total heat released, THR, or effective heat of combustion, EHC. In the case of the composite SR-3, a reduction of these parameters 60.7% (THR) and 66.7% (EHC) was obtained with regard to the reference composite SR-1 (Table 5, Figure 6).

At present, from the point of view of fire hazard, a lot of attention is being paid to the parameters directly reflecting the rate of fire development, i.e., tHRR_max_, AMLR (average mass loss rate of the sample with respect to its surface), FIGRA (fire growth rate) or MARHE (maximum average rate of heat emission).

The composites containing 30 parts by weight of both BFL (SR-3) and BFS (SR-4) are characterized by a considerably higher tHRR_max_ parameter as compared with the reference sample SR-1. The FIGRA parameter, determined as a quotient of the value of HRR_max_ and the time necessary to obtain the maximum heat release rate HRR_max_, which is a typical indicator illustrating the fire growth rate for a specific material exposed to a defined heat flux, in the case of composites containing both BFL and BFS assumes decisively lower values in comparison with the composite SR-1.

The data included in Table 5 clearly indicate that the value of the FIGRA parameter is strongly dependent on the amount of the basalt filler introduced into the SR rubber matrix.

The parameters AMLR and MARHE also indicate an effective flame retardant effect of the basalt filler.

It should be emphasized with clarity that slightly worse values of the fire hazard parameters of the composite containing BLS (SR-4), with reference to BFL (SR-3), are related to their larger size of fibers in comparison to basalt flakes, and thus their worse dispersion in the silicone rubber matrix (Figure 7). A more effective flame retardant activity of the basalt filler in the form of basalt flakes, BFL, is also related to the greater difficulty in the diffusion of liquid destruction products onto the surface of a burning composite. The products of thermal decomposition may move towards the flame only through little spaces between thermally stable basalt flakes dispersed in the polymer; the so-called channeling effect.

The flame retardant activity of the basalt filler is a result of both its high heat capacity and high thermal stability. Basalt, by absorbing heat radiation energy, acts as a thermal shield protecting the polymer against the subsequent reactions of degradation and thermal destruction, which is proved, among other things, by an increase in the value of the parameter time to ignition t_i_. A reduction of flammability of the composites of silicone rubber containing the basalt filler is, however, a result of the high thermal stability of basalt and its suitability for the formation of an insulating boundary layer, effectively reducing the flow of mass and energy between the sample and the flame.

The images illustrating the remains after the burning of the composites SR-1, SR-3 and SR-4, respectively, clearly indicate that under the white residue consisting of silica, there exists a homogenous basalt layer (Figure 8).

Reviewing the source literature on the subject leads to the conclusion that the forces of cohesive interactions between silica particles resulting from the decomposition of SR are insufficient for the formation of a condensed boundary layer [12,25]. This is due to the fact that the obtained silica is present in the form of dust, with a low barrier factor, which is additionally easy to remove from the surface of the boundary layer through a stream of gases in the combustion zone. Therefore, it should be unambiguously stated that the formed strong basalt layer is responsible for the reduced flammability of the SR rubber composites (Figure 8). 

The effect of the basalt filler is also very strongly marked in the case of ceramizable composites. It should be noted that the parameter HRR_max_ of the composite SR-8 was reduced by 45.4% in comparison to the reference ceramizable composite SR-5, while the parameter tHRR_max_ increased from 410 s in the case of the composite SR-5 to 990 s for the composite SR-8. It is worth emphasizing that an increase in the value of the parameter tHRR_max_ is proportional to the content of the basalt filler in the composite. Also, the other parameters of fire hazard, such as THR, EHC, AMLR, FIGRA, MARHE, are closely correlated with the amount of BFL in the SR rubber matrix (Table 6, Figure 9).

Functioning of basalt as a thermal shield is also very clearly marked in the case of ceramizable composites. The composite SR-5 undergoes ignition after 310 s; introducing only 15 parts by weight of basalt flakes into the silicone rubber matrix extends this time by as much as 109% to 650 s, and increasing BFL to 20 parts by weight results in an increase in the parameter t_i_ to a value of 1125 s. It is worth noting that, in accordance with the methodology for carrying out measurements, a sample that has not been ignited after a duration of 600 s should be considered to be non-flammable (Figure 9).

In the case of ceramizable composites, the basalt fibers, BFS, demonstrate practically the same flame retardant efficiency, as expressed by the parameters t_i_, t_f-o,_ HRR, HRR_max_, tHRR_max_, EHC, EHC_max_ or MLR, as basalt flakes. This is probably due to the better dispersion of basalt fibers, in the presence of mineral fillers (composite SR-9) than in the SR rubber matrix alone (composite SR-4) (Figure 10).

The lower values of the parameters HRR_max,_ THR and FIGRA for the composite SR-8 in comparison to SR-9 result from smaller sizes and, consequently, better distribution of BFL in relation to BFS (Figure 11).

### 3.4. Micromorphology of Composites Prior to the Ceramization Process

The micromorphology of composites containing basalt filler is very strongly dependent on its type. Basalt flakes, which are characterized by a size of under 10 µm, disperse very well in the silicone rubber matrix, which is confirmed by SEM images of the composite SR-3 as well as by the EDS analysis. Distribution mappings for silicon, oxygen, gold, aluminum or carbon indicate the homogeneous dispersion of basalt flakes in the SR rubber matrix (Figure 12).

Basalt fibers characterized by the length of 1.5 mm and a diameter of 13 µm are very clearly visible both in the SEM images and the accompanying EDS mappings of the composite SR-4 (Figure 13).

Ceramizable composites are generally characterized by a high degree of heterogeneity, which is a direct result of the large particle size of of mineral fillers, mainly calcium carbonate, zinc borate, glass frit, and also easily agglomerating silica. 

The SEM images of the initial ceramizable composite SR-5, before the ceramization process, indicate relatively homogeneous distribution of particular mineral components in the SR rubber matrix. This is confirmed by EDS mappings. The map for silicon distribution indicates that the silica contained in the composite does not exhibit an excessive tendency for agglomeration. It should also be clearly pointed out that the map for zinc distribution and that for carbon distribution indicate homogeneous dispersion and therefore the distribution in the SR rubber matrix, of zinc borate and melamine cyanurate, respectively. The maps for the distribution of calcium and aluminum indicate a slight agglomeration of calcium carbonate as well as aluminum oxide (Figure 14).

The introduction of the basalt filler, both in the form of basalt flakes and basalt fibers, does not significantly influence the dispersion and distribution methods of the particular mineral particles in the silicone rubber matrix. Good distribution of silica, melamine cyanurate, zinc borate or basalt flakes is accompanied by small agglomerates of both calcium carbonate and aluminum oxide (Figure 15).

### 3.5. Micromorphology of Composites after the Ceramization Process

The micromorphology of ceramizable composites was also analyzed after a ceramization process consisting of heating the sample from room temperature to a temperature of 950 °C. The SEM images of the reference ceramizable composite after the ceramization process indicate that the formed ceramic structure is, to a considerable degree, homogeneous. Even in the case of a high magnification of ×500, there is no clear separation of phases. However, further analysis of the formed ceramic layer with the use of EDS technology indicated the presence of interphase structures. It should be clearly pointed out that both on the silicon distribution map and on the oxygen distribution map for the composite SR-5, the silica agglomerates, which were previously absent, are clearly marked, and their formation is accompanied by the condensation of hydroxyl groups. Decisively larger aggregates of CaO and AlO in the ceramic layer, in comparison to the initial composite SR-5, confirm that calcium oxide and aluminum oxide, respectively, also undergo sintering. On the basis of the EDS mapping of zinc distribution it may be concluded that the zinc borate introduced into the silicone rubber matrix in the presence of glass frit undergoes flowing, combining more thermally stable particles of mineral fillers (Figure 16).

The introduction of basalt flakes does not significantly influence the image of ceramic structures obtained with the SEM method. Quite unexpectedly, however, on the basis of the EDS analyses (the distribution map for Si, Ca, Na, O), it was concluded that in the presence of basalt flakes, the degrees of aggregation of silica, calcium oxide, zinc borate and aluminum oxide are considerably lower. It is probably that the basalt filler, which is characterized by both high heat capacity and large particle size, reduces the processes of formation of so-called filler islands, thus contributing towards the formation of a homogeneous solid ceramic layer (Figure 17).

### 3.6. Mechanical Properties

The introduction of the basalt filler, both in the form of basalt flakes (BFL) and basalt fibers (BFS), into the silicone rubber matrix ambiguously influences the mechanical parameters of the obtained SR composites (Table 7, Figure 18). In the presence of BFL, in the case of the vulcanizates SR-1÷3, the value of the parameter TS clearly increases, which may indicate the formation of a secondary lattice of the filler, which strengthens the SR composite. At the same time, in the presence of BFL, the elasticity of the analyzed samples significantly decreases, which is indicated by a drop in the value of elongation at break, the parameter E_b_. It should also be noted that, in contrast to BFL, basalt fibers considerably worsen the mechanical parameters TS and E_b_ of composite SR-4 relative to the reference composite SR-0. This is, above all, related to the large size of the fiber and its poor distribution in the SR rubber matrix.

The basalt filler has practically no impact on the values of the parameters TS (MPa) and E_b_ (%) of ceramizable composites. In the presence of mineral fillers, which increase the distances between the basalt particles dispersed in the rubber matrix, no spatial structures are formed that could strengthen the studied ceramizable composites.

It should be noted that the basalt filler, both in the form of BFL and BFS, considerably strengthens the formed ceramic structure (Table 8). The composite SR-8 containing 30 parts by weight of BFL is characterized by a compressive strength almost twice as high as the reference composite (mild heating to T = 950 °C). The impact of the basalt filler on the durability of the obtained ceramic structure is also clearly marked while carrying out ceramization, taking into account the isothermal segment at T = 550 °C (Table 8, Figure 19).

It cannot be excluded that the basalt filler, due to its high thermal capacity, catalyzes the processes of melting and flowing of both zinc borate and glass frit, as well as the sintering of mineral fillers contained in the composite (Figure 17).

Extreme conditions of ceramization (heating to 1100 °C) result in the studied composites swelling strongly under the influence of the released gaseous products of thermal decomposition. However, the formed ‘honeycomb’-type ceramic structure is characterized by a very high degree of hardness. These samples could not be examined, owing to the limited measurement range of the durometer.

The obtained research results regarding compressive strength of the ceramic structure are compatible with the results obtained by Mansour et al. The ceramic layer formed as a result of the decomposition of ceramizable composites of SR rubber, at a temperature of T = 1000 °C, is characterized by a higher value of the force at bending than that obtained at T = 600 °C [27]. Similarly, Pędzich et al. discovered that the conditions of ceramization very strongly influence the microstructure of the ceramic layer, i.e., its porosity, pore size distribution, or homogeneity [28]. The samples of the ceramizable composite slowly heated to a temperature of T = 1000 °C form a ceramic layer of a homogeneous structure. The same samples heated rapidly to a temperature of T = 1000 °C develop a porous spongy ceramic structure.

## 4. Summary

An improvement of the thermal properties of the composites containing basalt is related to both the adsorption of polymer chains on the basalt surface and the high thermal capacity of basalt.

Increased polymer–filler interactions lead to a considerable reduction in the segment mobility of the polymer chains, and thus a reduction in the efficiency of the reaction of decomposition and chain transfer. In addition to this, basalt, by absorbing significant amounts of heat, acts as a thermal shield, which protects the composite against the processes of both degradation and destruction.

The reduction in the flammability of the silicone rubber composites containing basalt filler is, above all, a result of its high thermal stability and suitability for the formation of an insulating boundary layer that effectively reduces the flow of mass and energy between the sample and the flame.

The introduction of basalt flakes does not significantly influence the image of ceramic structures obtained using the SEM method. In the presence of basalt flakes, however, the degree of aggregation of silica, calcium oxide, zinc borate or aluminum oxide is considerably lower. It is probably that the basalt filler, which is characterized by high thermal capacity and large particle size, decreases the processes of the formation of the so-called filler islands, thus contributing towards the formation of a homogeneous solid ceramic layer, characterized by considerably better mechanical parameters in comparison to the reference composite.

## Figures and Tables

**Figure 1 materials-12-02432-f001:**
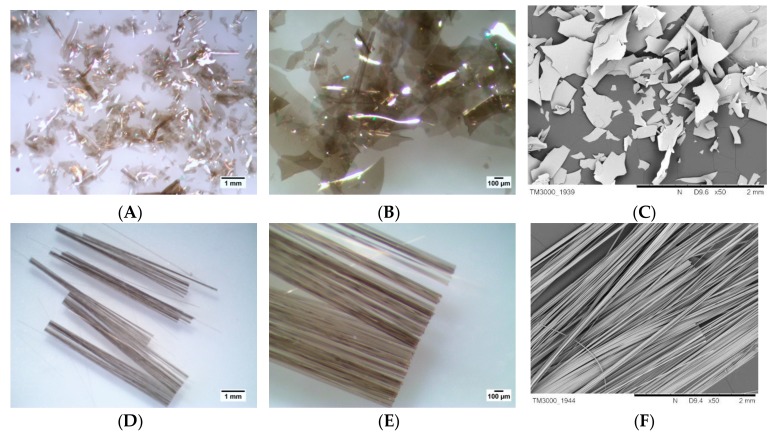
Basalt flake optic photos (**A**) ×10; (**B**) ×40. (**C**) Basalt flake SEM photo ×50. Basalt fiber optic photos (**D**) ×10; (**E**) ×40. (**F**) Basalt fiber SEM photo ×50.

**Figure 2 materials-12-02432-f002:**
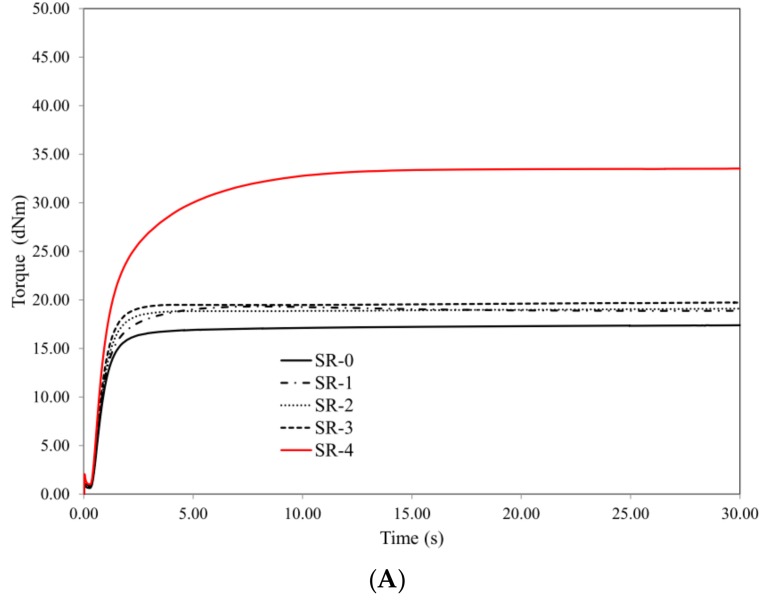
Vulcanization kinetics: (**A**) composite mixes, (**B**) ceramizable composite mixes.

**Figure 3 materials-12-02432-f003:**
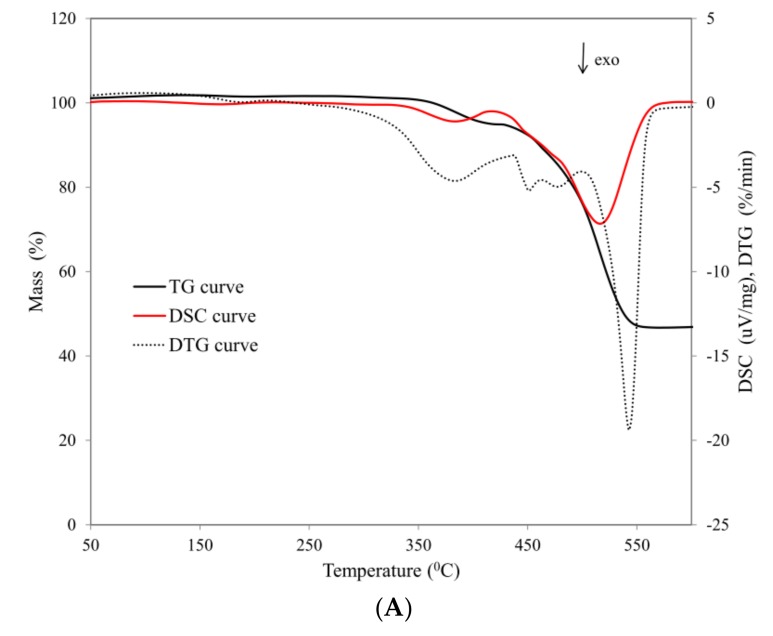
Thermal curves: (**A**) SR-0 composite (**B**) SR-5 composite.

**Figure 4 materials-12-02432-f004:**
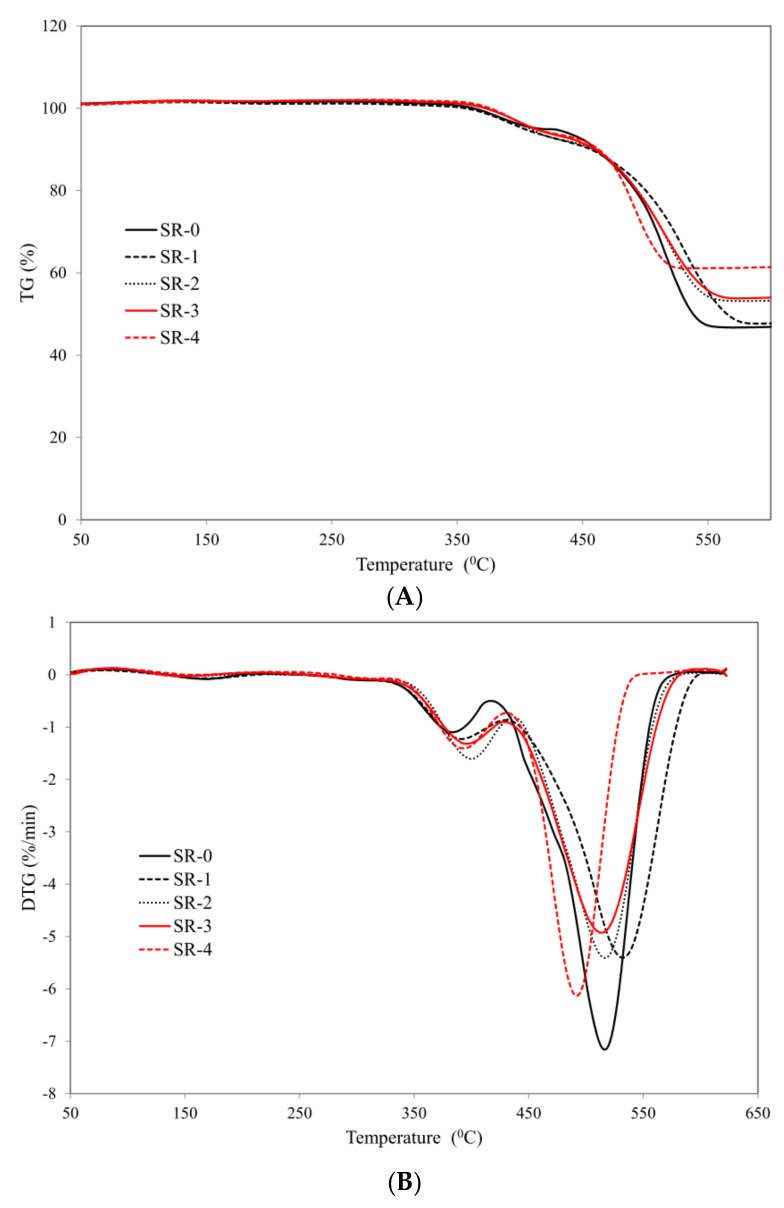
Thermal curves: (**A**) TG (**B**) DTG of studied composites.

**Figure 5 materials-12-02432-f005:**
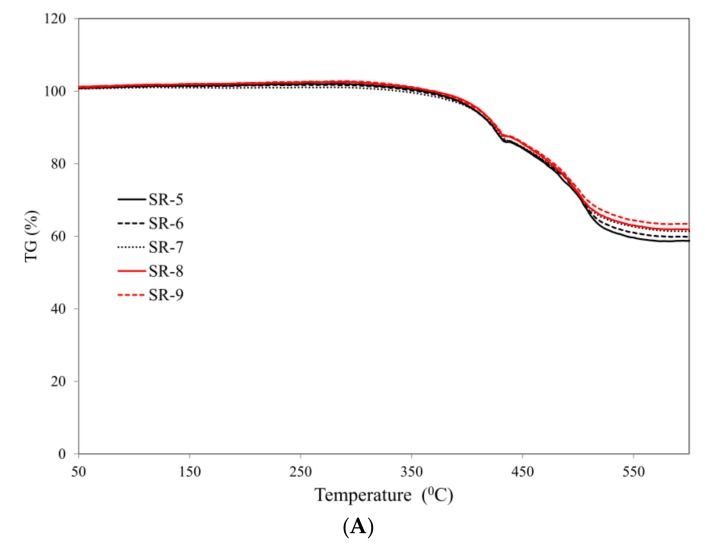
Curves: (**A**) TG (**B**) DTG of the studied ceramizable composites.

**Figure 6 materials-12-02432-f006:**
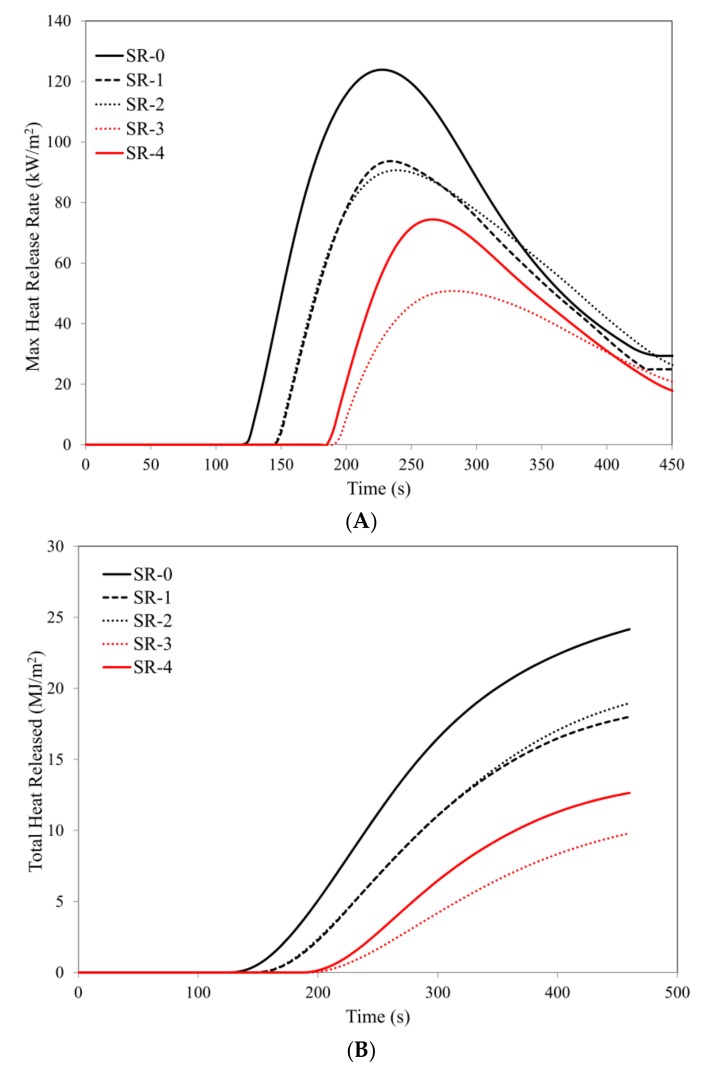
Cone calorimetry analysis: (**A**) max heat release rate, (**B**) total heat released, (**C**) average release rate (ARHE), (**D**) mass loss.

**Figure 7 materials-12-02432-f007:**
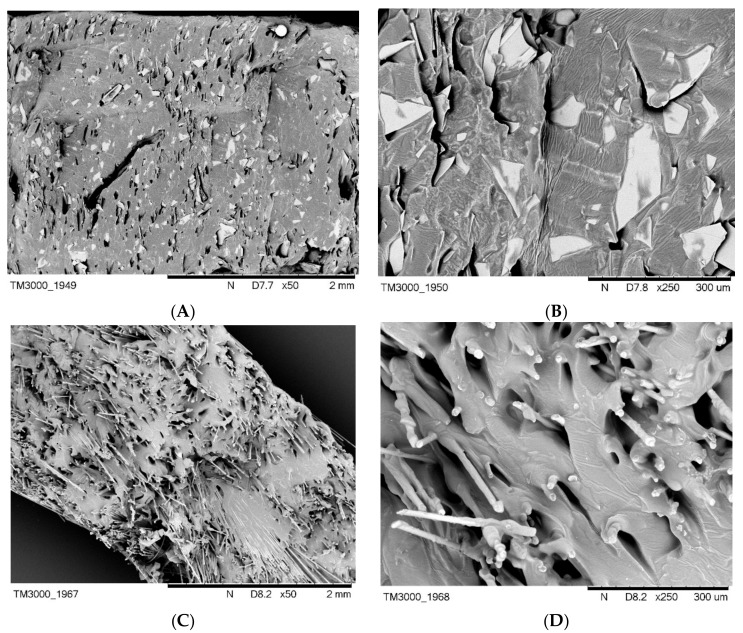
SEM images of non-vulcanized mixtures: (**A**) SR-3 ×50; (**B**) SR-3 ×250; (**C**) SR-4 ×50, (**D**) SR-4 ×250.

**Figure 8 materials-12-02432-f008:**
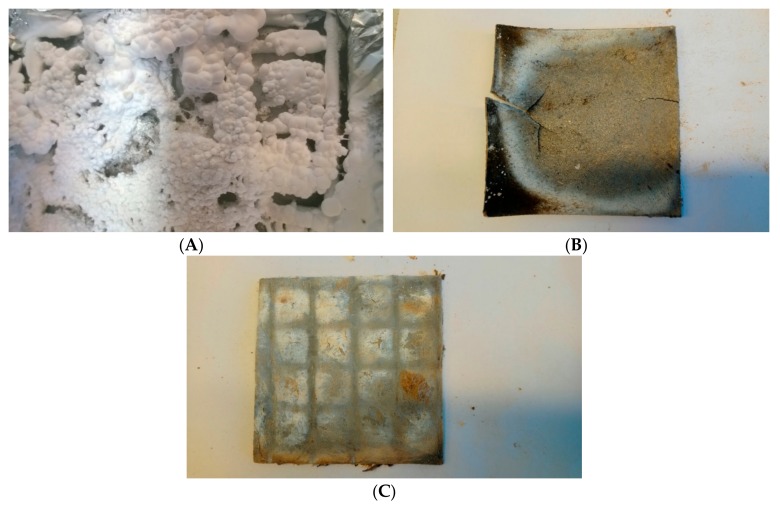
Remains after burning of the composites (**A**) SR-0, (**B**) SR-3 followed by removing the outer layer, and (**C**) SR-4 followed by removing the outer layer.

**Figure 9 materials-12-02432-f009:**
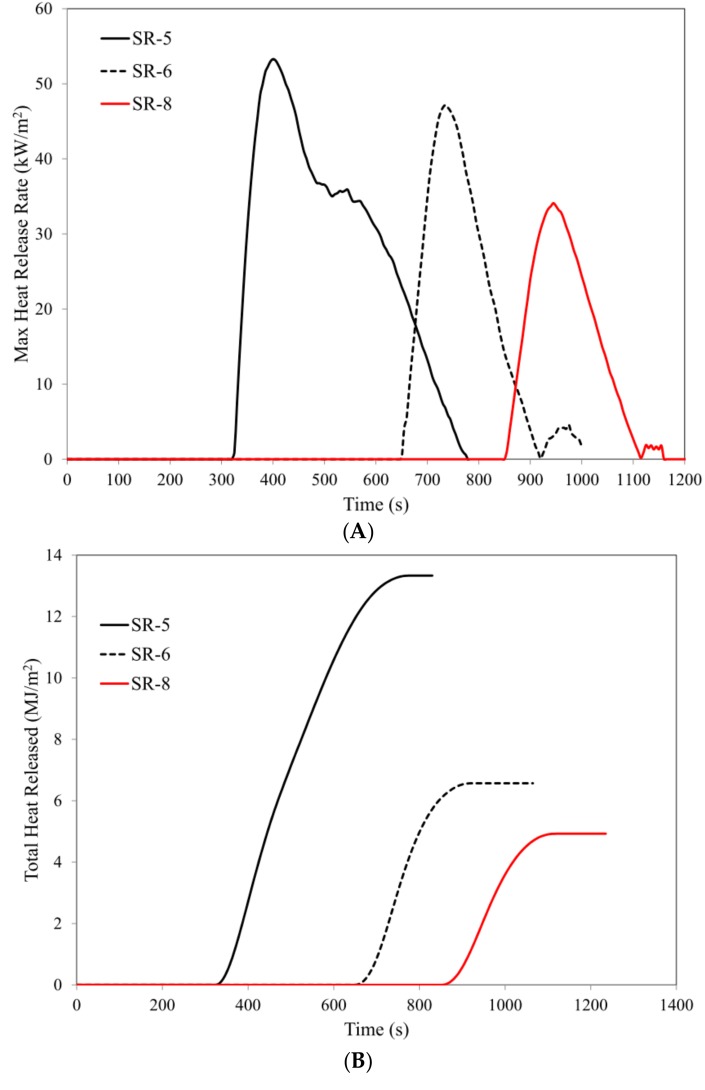
Cone calorimetry analysis: (**A**) max heat release rate, (**B**) total heat released, (**C**) average release rate (ARHE), (**D**) mass loss.

**Figure 10 materials-12-02432-f010:**
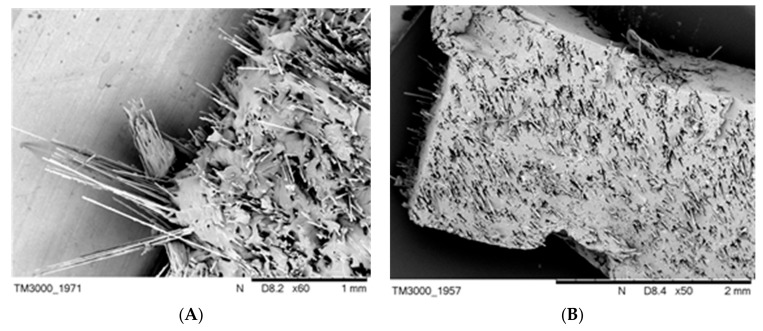
Images of elastomer mixes before vulcanization: (**A**) composite SR-3; (**B**) composite SR-9.

**Figure 11 materials-12-02432-f011:**
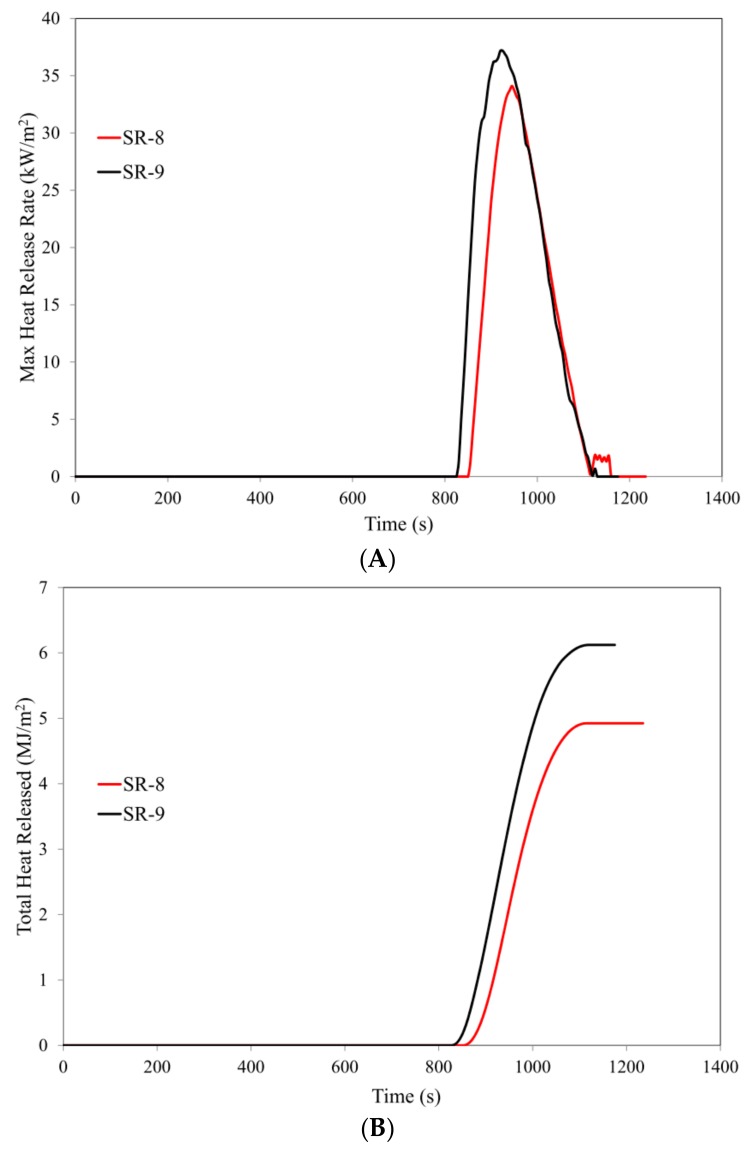
Cone calorimetry analysis: (**A**) max heat release rate, (**B**) total heat released, (**C**) average release rate (ARHE), (**D**) mass loss.

**Figure 12 materials-12-02432-f012:**
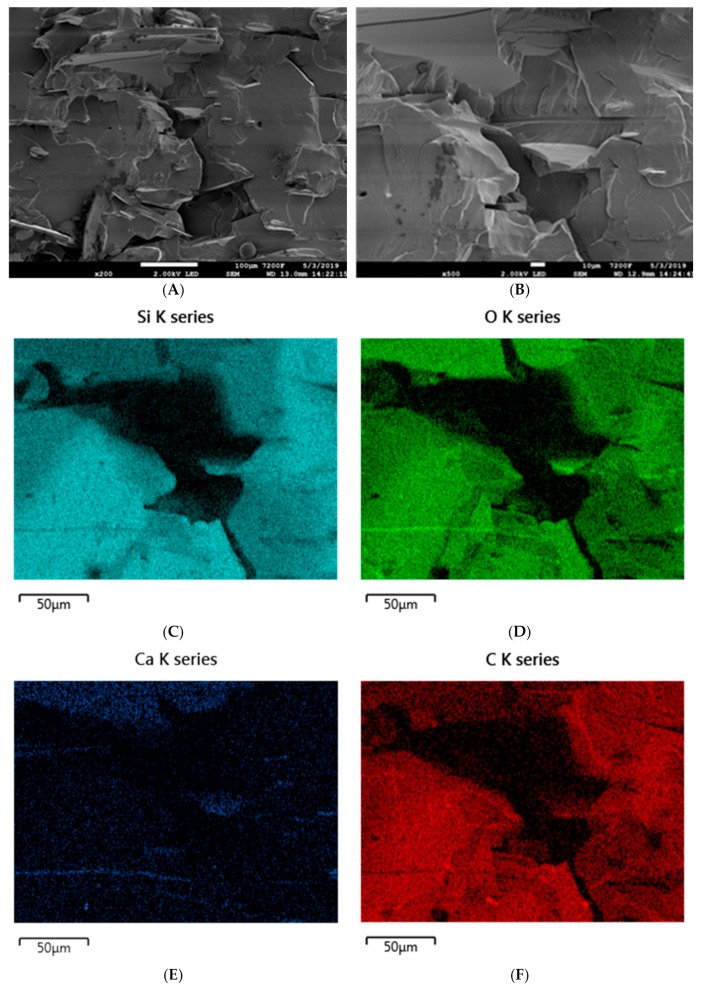
SEM images of the SR-3 composite at magnifications of (**A**) ×200 and (**B**) ×500. EDS mapping of SR-3 (**C**) silicon distribution, (**D**) oxygen distribution, (**E**) calcium distribution, and (**F**) carbon distribution.

**Figure 13 materials-12-02432-f013:**
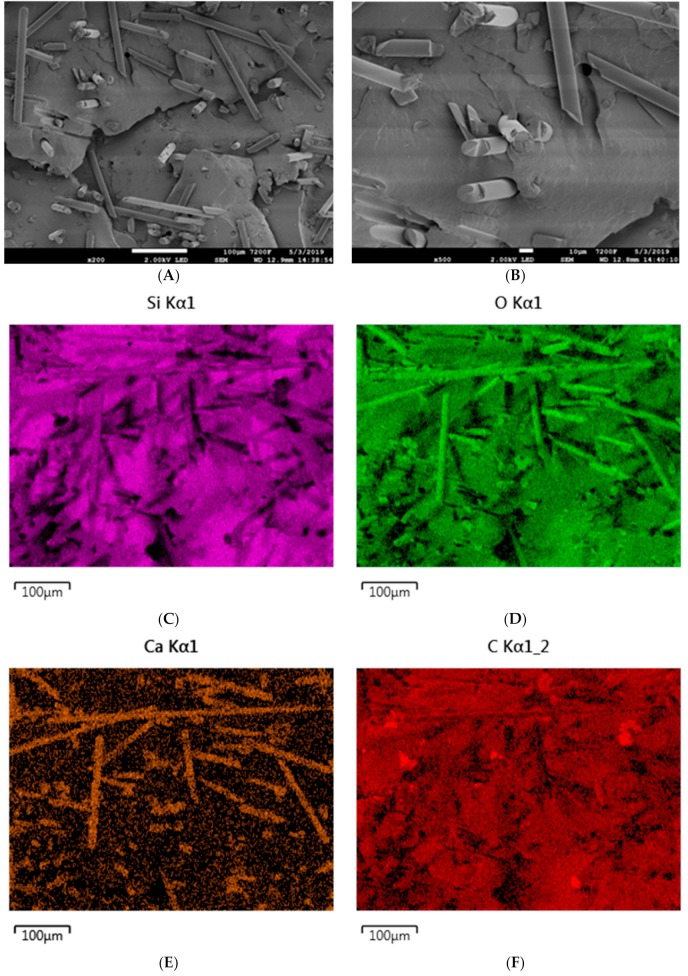
SEM images of the SR-4 composite at magnifications of (**A**) ×200 and (**B**) ×500. EDS mapping of SR-3 (**C**) silicon distribution, (**D**) oxygen distribution, (**E**) calcium distribution, (**F**) carbon distribution.

**Figure 14 materials-12-02432-f014:**
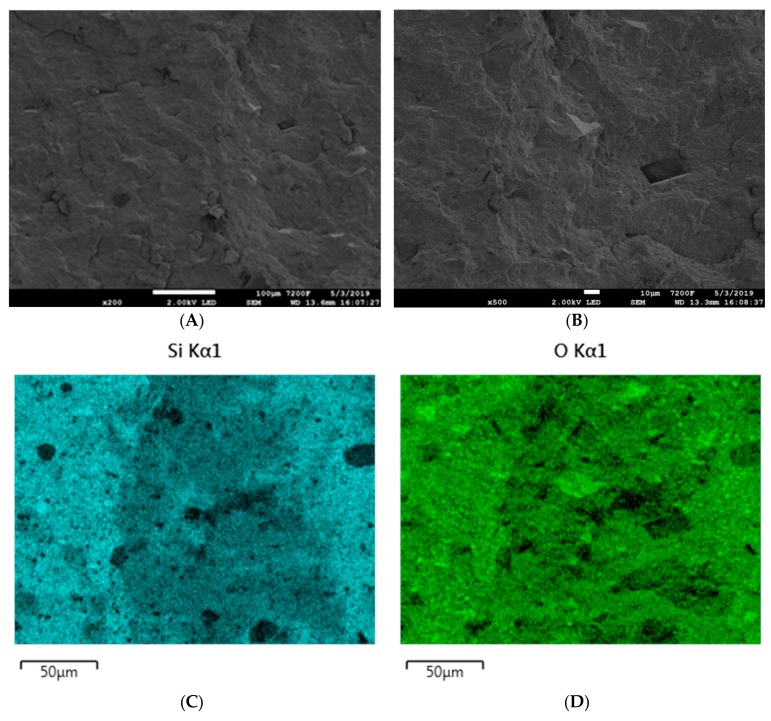
SEM images of the SR-5 composite at magnifications of (**A**) ×200 and (**B**) ×500. EDS mapping of SR-5 (**C**) silicon distribution, (**D**) oxygen distribution, € zinc distribution, (**F**) carbon distribution, (**G**) calcium distribution, and (**H**) aluminum distribution.

**Figure 15 materials-12-02432-f015:**
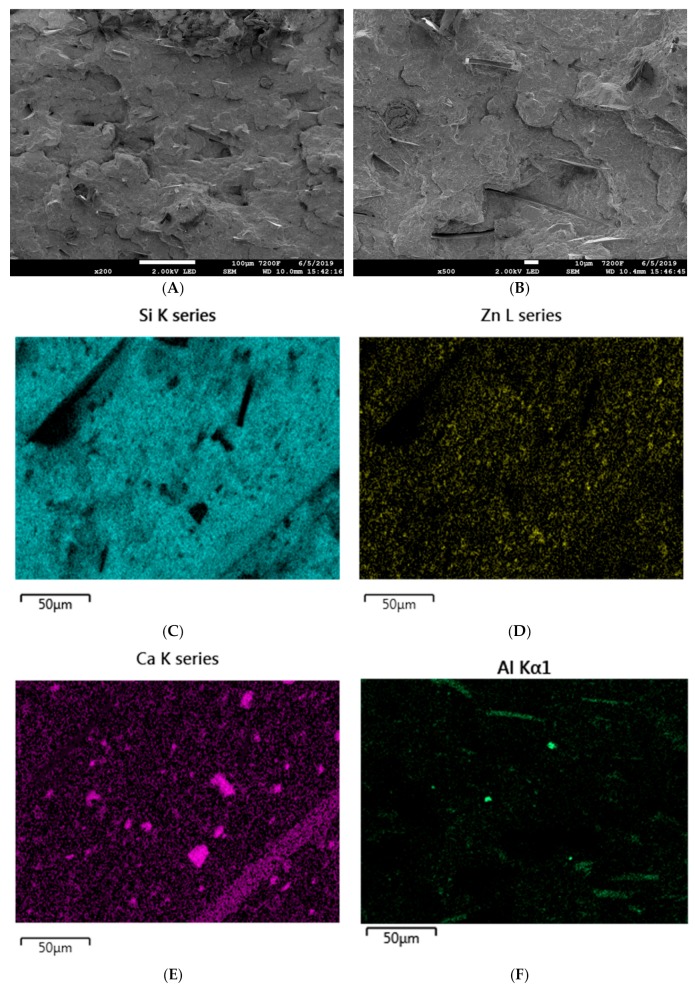
SEM images of the SR-6 composite at magnifications of (**A**) ×200 and (**B**) ×500. EDS mapping of SR-5 (**C**) silicon distribution, (**D**) zinc distribution, (**E**) calcium distribution, and (**F**) aluminum distribution.

**Figure 16 materials-12-02432-f016:**
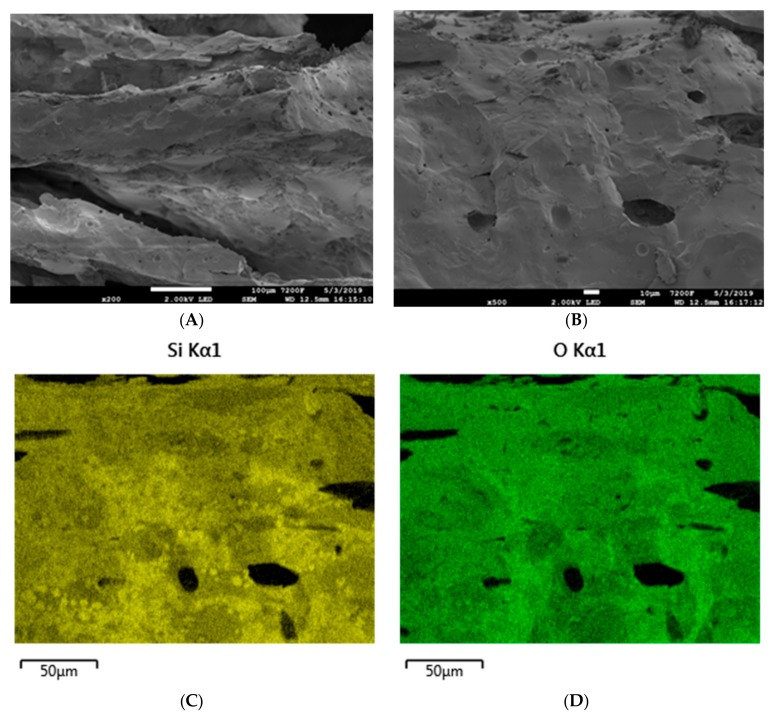
SEM images of SR-5 composite after its ceramization At a magnification of (**A**) ×200 and (**B**) ×500. EDS mapping of SR-5 (**C**) silicon distribution, (**D**) oxygen distribution, € calcium distribution, (**F**) aluminum distribution, (**G**) zinc distribution, and (**H**) calcium distribution.

**Figure 17 materials-12-02432-f017:**
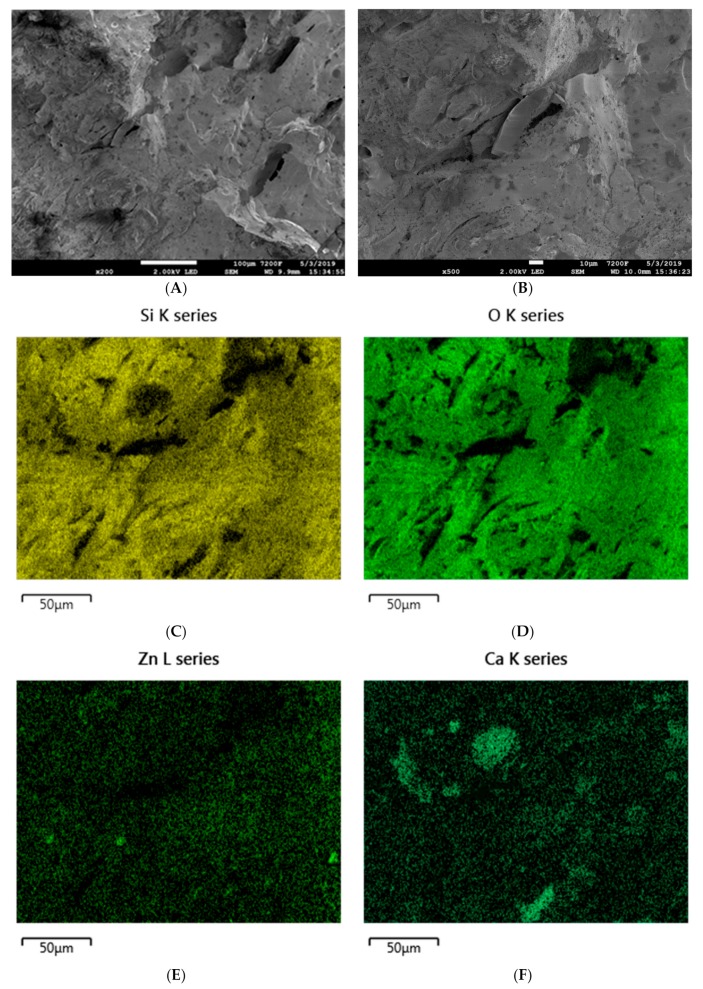
SEM images of the SR-6 composite after its ceramization At magnifications of (**A**) ×200 and (**B**) ×500. EDS mapping of SR-6 (**C**) silicon distribution, (**D**) oxygen distribution, (**E**) zinc distribution, and (**F**) calcium distribution.

**Figure 18 materials-12-02432-f018:**
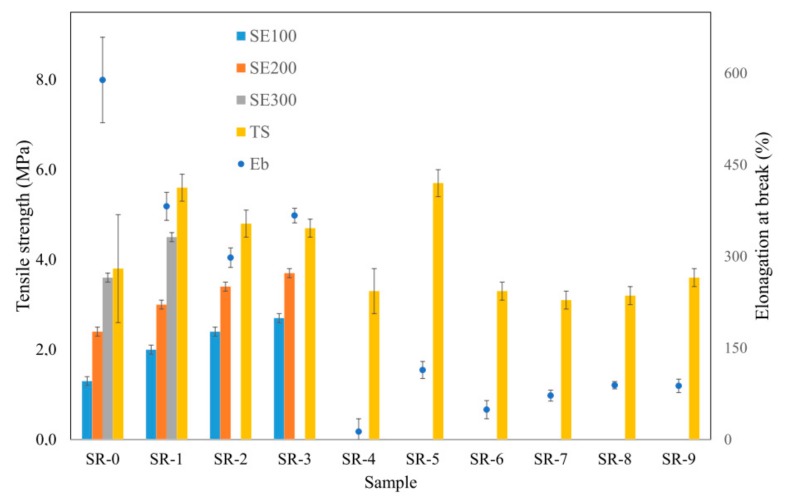
Mechanical properties of the studied composites.

**Figure 19 materials-12-02432-f019:**
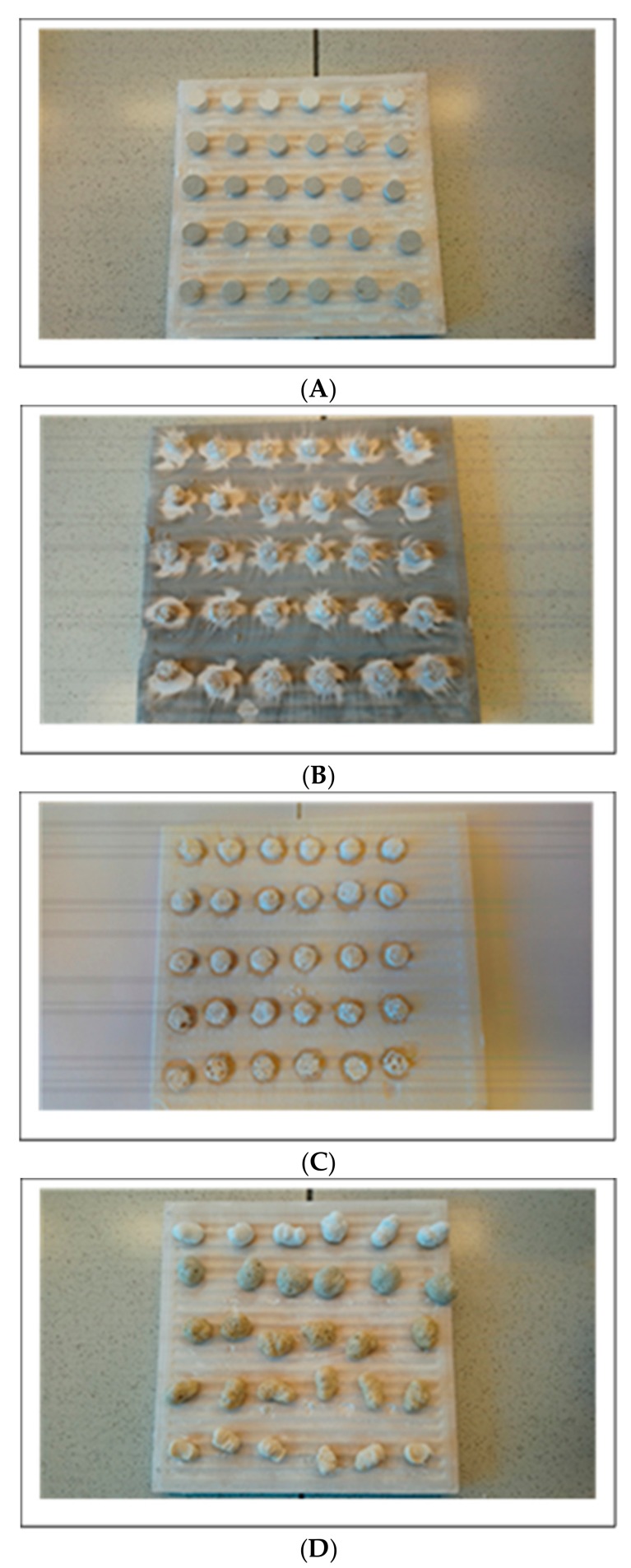
(**A**) Composites SR-5÷9 before ceramization, (**B**) Composites SR-5÷9 after ceramization at a temperature of 950 °C (programme 1), (**C**) Composites SR-5÷9 after ceramization at a temperature of 550–950 °C (programme 2), (**D**) Composites SR-5÷9 after ceramization at a temperature of 1100 °C (programme 3), (**E**) Composite SR-7 after ceramization at a temperature of T = 950 °C, (**F**) Composite SR-7 after ceramization at a temperature of T = 550–950 °C, (**G**) Composite SR-7 after ceramization at a temperature of T = 1100 °C.

**Table 1 materials-12-02432-t001:** Composition (in phr—parts per hundred parts of rubber) of the composite mixes.

Ingredient	Composition of the Samples (phr)				
	SR-0	SR-1	SR-2	SR-3	SR-4
SR	100	100	100	100	100
DCP	2.5	2.5	2.5	2.5	2.5
Silica	−	−	−	−	−
MC	−	−	−	−	−
CaCO_3_	−	−	−	−	−
Glass frit	−	−	−	−	−
ZnB	−	−	−	−	−
BFL	−	10	15	20	−
BFS	−	−	−	−	20

DCP—dicumyl peroxide; MC—melamine cyanurate; CaCO_3_—calcium carbonate; ZnB—zinc borate; BFL—basalt flakes; BFS—basalt fibers.

**Table 2 materials-12-02432-t002:** Composition (in phr—parts per hundred parts of rubber) of the ceramizable composite mixes.

Ingredient	Composition of the Samples (phr)				
	SR-5	SR-6	SR-7	SR-8	SR-9
SR	100	100	100	100	100
DCP	2.5	2.5	2.5	2.5	2.5
Silica	15	30	15	15	15
MC	30	30	30	30	30
CaCO_3_	15	15	15	15	15
Glass frits	30	30	30	30	30
ZnB	15	15	15	15	15
BFL	−	10	15	20	−
BFS	−	−	−	−	20

**Table 3 materials-12-02432-t003:** Vulcanization parameters of the composite mixes.

Vulcanized Composites Description	Vulcanization Parameters			
	Scorch Time t_05_ (s)	Torque at t_05_ (dNm)	Optimum Curing Time t_90_ (s)	Torque at t_09_ (dNm)
SR-0	25	0.62	113	14.89
SR-1	25	0.75	141	17.21
SR-2	25	0.76	104	16.11
SR-3	25	0.82	99	16.72
SR-4	26	0.99	316	30.12
SR-5	24	5.21	306	35.82
SR-6	25	6.21	180	38.74
SR-7	24	6.13	126	37.82
SR-8	24	6.17	124	38.01
SR-9	24	4.73	183	37.11

**Table 4 materials-12-02432-t004:** Thermal properties of SR rubber composites.

Vulcanized Composites Description	Thermal Properties Parameter					
	T_5_ (°C)	T_50_ (°C)	T_rmax1_ (°C)	T_rmax2_ (°C)	dm/dt (%/min)	P_650_ (%)
SR-0	415	538	383	516	7.15	46.8
SR-1	402	560	390	535	5.96	47.7
SR-2	411	−	398	518	5.38	54.0
SR-3	408	−	397	516	5.41	53.2
SR-4	412	−	391	492	6.13	61.4
SR-5	403	−	424	500	3.48	58.7
SR-6	404	−	423	497	3.29	61.3
SR-7	404	−	423	500	3.38	59.8
SR-8	409	−	421	494	3.21	61.9
SR-9	408	−	424	494	3.01	63.4

T_5_, T_50_—temperature of the sample at 5% and 50% mass loss, respectively; T_rmax_—temperature of maximum rate of thermal decomposition of composites; dm/dt—maximum rate of thermal decomposition of vulcanizates; P_650_—residue after heating to T = 650 °C.

**Table 5 materials-12-02432-t005:** Flammability of composites with BFL and BFS.

Combustibility Parameters	Vulcanized Composites Description				
	SR-0	SR-1	SR-2	SR-3	SR-4
t_i_ (s)	115	145	147	190	195
t_f-o_ (s)	390	690	710	730	725
HRR (kW/m^2^)	76.6	31.7	36.2	15.3	25.3
HRR_max_ (kW/m^2^)	134.7	96.2	90.7	48.5	77.3
tHRR_max_ (s)	225	230	245	290	270
THR (MJ/m^2^)	24.7	17.1	19.3	9.7	12.5
EHC (MJ/kg)	38.2	23.9	25.1	12.7	19.2
EHC_max_ (MJ/kg)	80.8	78.5	78.1	68.2	79.9
MLR (g/s)	0.02	0.013	0.013	0.01	0.011
MLR_max_ (g/s)	0.07	0.044	0.043	0.044	0.046
AMLR (g/m^2^ × s)	2.89	2.82	2.30	1.43	1.44
FIGRA (kW/m^2^s)	0.59	0.41	0.37	0.16	0.28
MARHE (kW/m^2^)	67.3	43.1	41.5	19.3	30.3
Burning droplets	no	no	no	no	no

t_i_—time to ignition; t_f-o_—time to flameout; HRR—heat release rate; HRR_max_—maximum heat release rate; tHRR_max_—time to maximum heat release rate; THR—total heat release; EHC—effective heat of combustion; EHC_max_—maximum effective heat of combustion; MLR—mass loss rate; MLR_max_—maximum mass loss rate; AMLR—average mass loss rate; FIGRA—HRR_max_/tHRR_max_; MARHE—maximum average heat of emission.

**Table 6 materials-12-02432-t006:** Flammability of ceramizable composites.

Combustibility Parameters	Vulcanized Composites Description				
	SR-5	SR-6	SR-7	SR-8	SR-9
t_i_ (s)	310	650	665	850	819
t_f-o_ (s)	780	920	990	1125	1120
HRR (kW/m^2^)	27.3	20.1	21.4	14.9	17.0
HRR_max_ (kW/m^2^)	55.2	48.3	43.4	33.1	37.9
tHRR_max_ (s)	410	770	780	990	930
THR (MJ/m^2^)	14.4	7.6	6.9	4.7	6.3
EHC (MJ/kg)	16.7	16.3	14.5	11.8	13.9
EHC_max_ (MJ/kg)	57.1	43.6	41.1	47.5	43.5
MLR (g/s)	0.01	0.01	0.01	0.01	0.01
MLR_max_ (g/s)	0.05	0.05	0.042	0.022	0.029
AMLR (g/m^2^ × s)	1.21	1.11	1.01	0.89	0.89
FIGRA (kW/m^2^s)	0.13	0.06	0.05	0.03	0.04
MARHE (kW/m^2^)	18.5	7.3	6.38	4.51	5.62
Burning droplets	no	no	no	no	no

**Table 7 materials-12-02432-t007:** Mechanical properties of the studied composites.

Mechanical Parameter	Vulcanized Composites Description									
	SR-0	SR-1	SR-2	SR-3	SR-4	SR-5	SR-6	SR-7	SR-8	SR-9
SE100 (MPa)	1.3 ± 0.1	2.0 ± 0.1	2.4 ± 0.1	2.7 ± 0.1	−	−	−	−	−	−
SE200 (MPa)	2.4 ± 0.1	3.0 ± 0.1	3.4 ± 0.1	3.7 ± 0.1	−	−	−	−	−	−
SE300 (MPa)	3.6 ± 0.1	4.5 ± 0.1	−	−	−	−	−	−	−	−
TS (MPa)	3.8 ± 1.2	5.6 ± 0.3	4.8 ± 0.3	4.7 ± 0.2	3.3 ± 0.2	3.7 ± 0.5	3.3 ± 0.3	3.1 ± 0.2	3.2 ± 0.2	3.6 ± 0.2
E_b_ (%)	589 ± 70	382 ± 23	298 ± 16	367 ± 12	114 ± 11	49 ± 21	72 ± 14	89 ± 15	88 ± 9	13 ± 6

SE100, 200, 300 stress at 100, 200, 300% of elongation; TS—tensile strength; E_b_—elongation at break.

**Table 8 materials-12-02432-t008:** Compression strength of the ceramized studied composites.

Ceramization Conditions	Vulcanized Composites Compression Strength (N)				
	SR-5	SR-6	SR-7	SR-8	SR-9
950 °C	645 ± 155	710 ± 130	898 ± 263	1263 ± 168	937 ± 176
550–1000 °C	729 ± 11	832 ± 265	968 ± 191	1016 ± 265	1000 ± 23
1100 °C	−	−	−	−	−

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
