# Peer review of "Impact of Basalt Filler on Thermal and Mechanical Properties, as Well as Fire Hazard, of Silicone Rubber Composites, Including Ceramizable Composites"

_materials, 2019, doi:10.3390/ma12152432_

Round 1
Reviewer 1 Report
This is an interesting paper on the effect of basalt filler addition on the thermal, mechanical and fire resistance properties of silicone rubber composites. The paper is overall fine. The following points may help the authors to further improve their manuscript:
- Lines 22-23, please remove the highlight text
- Lines 29-30, 75-79, please rephrase
- Please provide details on the mechanical testing procedure. Did you follow any standard methodology? What was the size of the specimens? Under what speed was testing performed? What was the load cell used?
- Figure 2: I suggest to use the same max value (i.e. 50) in both Fig 2a and 2b for direct comparison
- Figure 3: x-axis is not properly visible
- Please elucidate on SEM images presented in Fig. 7 and 10. The obtained morphology should be discussed.
- Add reference in statement 411-413
- Figures 12-17 do not add much in their current form. I suggest to reconsider the way these results are presented. In my opinion less images could convey the overall message
Author Response
Responses to the reviewers comments and improved version of the manuscript I send as an enclosures.
Reviewer 2 Report
In the manuscript (materials-537398), the authors verified the positive impact of basalt fillers on the thermal stability, fire safety and mechanical property of silicon rubber and ceramizable silicon rubber. Actually, the ceramization of silicon rubber exploited a feasible route for the electronic and fire safety. The idea to use the basalt fillers to improve the ceramization of SR was acceptable and appreciated. Even though the results and discussions were impressive, some questionable analysis was present. In terms of the ceramization process, the authors observed the morphology and tested the mechanical property before and after. Actually, the composition evolution was more critical in scientific viewpoint (ACS Appl. Mater. Interfaces, 2019, 11, 7459-7471). According to the above assessment, the reviewer suggested a major revision for publication in Materials. The detailed comments and questions were listed below;
1) In Abstract, Page 1, the authors mentioned the “rheological property” However, the authors did not analyze the rheological property. Also, in Abstract and Introduction, Page 1-2, the authors mentioned the durability parameters, however, the authors failed to analyze the parameters. A full check from the authors was lacking before the first submission.
2) In Introduction, Page 1-2, the authors used a lot of “The review of literature on the subject”. It was not scientific.
3) In Section 2.2, the molecular weight of SR was labeled as an incorrect unit.
4) Serious problem. In Section 2.3, Page 4, the authors used the power of 35kW/m2, which generated the temperature of roughly 682oC. This temperature was not able to reflect the positive effect of ceramization (1000oC).
5) In Section 3.1, Page 5, the vulcanization property should be re-written for a more full description. Please also check the labels (such as SR-5÷SR-9).
6) In Section 3.2, Page 7, the authors mentioned “which is directly related to the formation of smaller amounts of volatile - including flammable ones - products of pyrolysis reaching the flame, which in turn inhibits free radical reactions taking place in the combustion zone” and “An increase in the polymer-filler interactions results in a considerably lowered segment mobility of polymer chains, and therefore reduced efficiency of the decomposition reaction and the chain transfer.” The analysis was incorrect. Please check!
7) The language of this manuscript should be seriously modified. The good sentence should be simple but informative. Herein, the sentences were indeed difficult to understand even though what to be described was so simple.
8) In Section 3.3, Table 5, Page 8, the fire data should not be retained in two decimal places. Please revise them.
9) In Fig.6, the HRR curves in cone calorimetry were not performed in a correct manner. The baseline and stop point during operation was not selected properly.
10) In Conclusion, Page 25, the authors mentioned “Increased polymer-filler interactions lead to a considerable reduction of segment mobility of polymer chains, and thus a reduction in the efficiency of the reaction of decomposition and chain transfer. Apart from this, basalt, by absorbing significant amounts of heat, acts as a thermal shield which protects the composite against the processes of both degradation and destruction. The reduction of the flammability of silicone rubber composites containing the basalt filler is above all a result of its high thermal stability and suitability for the formation of an insulating boundary layer which effectively reduces the flow of mass and energy between the sample and the flame.” Please vigorously revise it.
11) Excessive figures were present in the manuscript. Move the less important to Supporting Information.
Author Response

(The authors gave the same response as above.)

Reviewer 3 Report
This manuscript discusses the basalt filler’s impact on silicone rubber’s mechanical, thermal, and fire performance. A wide range of tests have been performed, and a lot of standard tests were performed to analyze those properties. However, the length of this manuscript is excessively long and contains a lot of redundant information. The paper is poorly formatted and not carefully written. Some problems are:
1) Line 99, “cross-linked with bis 2,4 dichlorobenzoil peroxide”. This contradicts to the dicumyl peroxide in Table 1 and 2.
2) Mixed use of decimal points (“,” and “.”) in tables and figures.
3) Table 1 and 2. The composition needs to be normalized by the total weight otherwise hard to compare. It is also very important to note that in your samples, the inert materials composition varies. Some effect on the thermal/fire performance might just be the physical additive effect.
4) Figures 2-4, 9 are poorly cropped and some of the information is missing. Some figures are blurring.
5) Figure 6, I don’t think you can plot the max heat release rate. The maximum heat release rate should be a single value (maximum one) from the heat release rate history. The heat release rate shows negative values of -40 kW/m2. This is not physically possible unless you have significant amount of oxygen released before ignition. Otherwise it should be something related to the instrument problem. The average release rate (ARHE) looks strange. If the authors use a new parameter, they need to define that carefully with math expressions and show it is more meaningful than some traditional parameters (i.e. peak heat release rate).
6) Figure 9 and 11. Similar problem with the negative (max) heat release rate. Also, SR-7 information is missing.
Author Response

(The authors gave the same response as above.)

Round 2
Reviewer 2 Report
In the response letter and revised manuscript (materials-537398-V2), the reviewer did not obtain the satisfied response related to the critical comment and question. In consideration of the high requirement of the scientific spirit, the reviewer persisted the major revision before reconsideration for publication in the Journal Materials.
1) Question 1. In the rheology section, the term “rheology” should be change into the “processing rheology” for clear understanding. In the normal sense, the rheology should be divided to the structural rheology and processing rheology, which were respectively corresponded to the low and high shear rate. Without the specific annotation, the rheology was considered to behave in the oscillatory rheometer.
2) Question 2. In terms of the molecular weight of the silicon rubber, the authors would like to know the reason why the SR with so high molecular weight (108g/mol) was capable to keep viscous at ambient temperature. The reviewer did not want the technical sheet. Please comment carefully.
3) Question 3. In terms of the temperature of the sample surface, the reviewer was confident that temperature merely kept at most 500oC during combustion. The reviewer ever tested the surface temperature at different powers. The flame temperature and the sample temperature were two things. The reviewer strongly suggested showing a persuasive comment to this point.
4) Question 5. The reviewer firmly persisted that there were no definite relation between dm/dt and flammability (LOI and UL-94) due to the influence of sample composition and dimensions. Additionally, there was not research to demonstrate the clear relation between segment mobility and thermal stability. Segment mobility and thermal stability originated from the molecule structures. Therefore, it was acceptable that the specific molecule structure resulted in the higher thermal stability. Also heat penetration and molecule break acted as the bridge between molecule structure and thermal stability. Please check.
5) Question 9. Indeed, the reviewer persisted that the baseline problem was present. As the research was interesting and the creditability of fire data, the reviewer suggested showing some persuasive comments in the manuscript rather than repeating these samples.
Author Response
Responses for reviewers comments we send as an enclosure.

Reviewer 3 Report
The cone calorimeter is based on a the oxygen consumption principle. There is no chimney effect evolved. The duct flow rate is accurately controlled and constantly measured from the pressure transducer. We have dealt with cone calorimeter for many years (also with FTT) and have never seen any very high negative HRR in any tests. Your HRR tests results are possibly problematic with a faulty instrument.
The cone calorimeter test is a standard test in many standards (ISO, ASTM). The calculation of essential parameters are strictly defined. I can understand some cone manufacturer try to get some extra “fancy” parameters which is basically from the same heat release rate (HRR) but follows different algorithms. Different manufacturer might give different names. The definition or math expression of many of these "extra" parameters is not rigorously justified.
Author Response

(The authors gave the same response as above.)
